# β2 Integrins on Dendritic Cells Modulate Cytokine Signaling and Inflammation-Associated Gene Expression, and Are Required for Induction of Autoimmune Encephalomyelitis

**DOI:** 10.3390/cells11142188

**Published:** 2022-07-13

**Authors:** Monika Bednarczyk, Vanessa Bolduan, Maximilian Haist, Henner Stege, Christoph Hieber, Lisa Johann, Carsten Schelmbauer, Michaela Blanfeld, Khalad Karram, Jenny Schunke, Tanja Klaus, Ingrid Tubbe, Evelyn Montermann, Nadine Röhrig, Maike Hartmann, Jana Schlosser, Tobias Bopp, Björn E Clausen, Ari Waisman, Matthias Bros, Stephan Grabbe

**Affiliations:** 1Department of Dermatology, University Medical Center, Johannes Gutenberg-University Mainz, Langenbeckstraße 1, 55131 Mainz, Germany; m.bednarczyk.09@aberdeen.ac.uk (M.B.); vbolduan@students.uni-mainz.de (V.B.); mhaist@uni-mainz.de (M.H.); henner.stege@unimedizin-mainz.de (H.S.); c.hieber@imb-mainz.de (C.H.); jschunke@students.uni-mainz.de (J.S.); tklaus@students.uni-mainz.de (T.K.); tubbe@uni-mainz.de (I.T.); monterma@uni-mainz.de (E.M.); n.roehrig@uni-mainz.de (N.R.); maike.hartmann@unimedizin-mainz.de (M.H.); schlosser.jana@stud.hs-fresenius.de (J.S.); mbros@uni-mainz.de (M.B.); 2Institute for Molecular Medicine, University Medical Center, Johannes Gutenberg University of Mainz, Langenbeckstraße 1, 55131 Mainz, Germany; lijohann@students.uni-mainz.de (L.J.); cschelmb@uni-mainz.de (C.S.); blanfeld@uni-mainz.de (M.B.); karram@uni-mainz.de (K.K.); bclausen@uni-mainz.de (B.E.C.); waisman@uni-mainz.de (A.W.); 3Research Center for Immunotherapy (FZI), University Medical Center, Johannes Gutenberg University of Mainz, Langenbeckstraße 1, 55131 Mainz, Germany; boppt@uni-mainz.de; 4Institute of Immunology, University Medical Center, Johannes Gutenberg University Mainz, Langenbeckstraße 1, 55131 Mainz, Germany

**Keywords:** β2 integrins, dendritic cells, cytokine, signal transducers and activators of transcription, suppressor of cytokines, experimental autoimmune encephalomyelitis

## Abstract

Heterodimeric β2 integrin surface receptors (CD11a-d/CD18) are specifically expressed by leukocytes that contribute to pathogen uptake, cell migration, immunological synapse formation and cell signaling. In humans, the loss of CD18 expression results in leukocyte adhesion deficiency syndrome (LAD-)1, largely characterized by recurrent severe infections. All available mouse models display the constitutive and ubiquitous knockout of either α or the common β2 (CD18) subunit, which hampers the analysis of the cell type-specific role of β2 integrins in vivo. To overcome this limitation, we generated a CD18 gene floxed mouse strain. Offspring generated from crossing with CD11c-Cre mice displayed the efficient knockdown of β2 integrins, specifically in dendritic cells (DCs). Stimulated β2-integrin-deficient splenic DCs showed enhanced cytokine production and the concomitantly elevated activity of signal transducers and activators of transcription (STAT) 1, 3 and 5, as well as the impaired expression of suppressor of cytokine signaling (SOCS) 2–6 as assessed in bone marrow-derived (BM) DCs. Paradoxically, these BMDCs also showed the attenuated expression of genes involved in inflammatory signaling. In line, in experimental autoimmune encephalomyelitis mice with a conditional DC-specific β2 integrin knockdown presented with a delayed onset and milder course of disease, associated with lower frequencies of T helper cell populations (Th)1/Th17 in the inflamed spinal cord. Altogether, our mouse model may prove to be a valuable tool to study the leukocyte-specific functions of β2 integrins in vivo.

## 1. Introduction

β2 integrins constitute heterodimeric surface receptors that consist of a variable α subunit (CD11a-CD11d) and a constant β subunit (CD18). They are exclusively expressed by leukocytes [1]. Lymphocyte function-associated antigen (LFA-1, CD11a/CD18) is the only β2 integrin that is expressed by all types of leukocytes. LFA-1 and macrophage-1 antigen (MAC-1, CD11b/CD18), which is predominantly apparent on myeloid cells [2] including conventional dendritic cells (cDCs) [3,4], confer cell–cell interactions by binding intercellular adhesion molecules (ICAMs) [5]. Such cellular interactions are of pivotal importance in the context of T cell activation, which requires the formation of an immunological synapse (IS) between antigen-presenting cells (APCs) such as DCs and T cells [6]. The interaction between β2 integrins and ICAM is a prerequisite for IS formation, as delineated for LFA-1 [7] and MAC-1 [8]. Furthermore, we previously reported that LFA-1 [9] and MAC-1 [10] on DCs, as well as LFA-1 on T cells [11], also regulate the extent of CD4^+^ T cell activation and their polarization. In addition, LFA-1 and MAC-1 play important roles in lymphocyte and granulocyte migration, respectively, from the blood into inflamed tissues [12].

Furthermore, MAC-1 and CD11c/CD18, which constitutes a pan-DC marker in mice [13], serve as complement receptors that recognize complement-opsonized immune complexes and pathogens, as well as mediate their uptake [14]. Additionally, MAC-1 binds a large variety of extracellular matrix proteins and serum factors, conferring their endocytosis [15]. MAC-1 also constitutes the main recognition and phagocytic receptor for complement-opsonized pathogens on myeloid cells and natural killer (NK) cells [16]. Interestingly, MAC-1 acts as a co-receptor for the phagocytosis of antibody-opsonized particles by Fc receptors [17], and (besides phagocytosis [18]) has been shown to regulate various pathogen defense mechanisms in myeloid cells such as reactive oxygen species production and NETosis by polymorphonuclear leukocytes (PMNs) [19,20]. MAC-1 has further been demonstrated to act as a signaling adaptor, fine-tuning the extent of the activation of monocytic cells [21] and DCs [22] in response to stimulation. Under homeostatic conditions in most cell types, β2 integrins display a low affinity state [23] that is enhanced in response to activation by conformational changes of the otherwise closed headpiece [24]. Activation may be conferred by binding of ligands or counter-receptors (“outside–in signaling”) and in response to activation via distinct receptors such as chemokines, which trigger intracellular signals (“inside–out signaling”) [25].

In humans, the loss-of-function mutation of CD18 results in leukocyte adhesion deficiency syndrome (LAD-)1, which is largely characterized by the frequent occurrence of severe infections [26]. This phenotype has mainly been attributed to the functional impairment of PMNs and other innate myeloid cell types to immigrate into infected sites and to kill pathogens [27]. Notably, LAD1 patients have been reported to be at higher risk for autoimmune disease [28], which suggests an attenuated maintenance of T cell tolerance in the periphery that is predominantly mediated by DCs [29].

So far, the cell type-specific role of β2 integrins for immune cell functions has mainly been studied using mouse strains with ubiquitously impaired expression (CD18^hypo^) [30], deficiency (CD18^−/−^) [31] or the expression of a mutated form [32] of the constant β subunit CD18 or either α subunit (CD11a [33], CD11b [34], CD11c [35], and CD11d [36]) in a variety of disease models. However, due to the unspecific character of β2 integrin knock-down/deficiency and the extensive interaction of leukocyte populations, it is difficult to ascribe β2 integrins cell type-specific functions in vivo.

Here we describe the generation of a mouse strain with a floxed CD18 gene locus, allowing for the knockdown of β2 integrins in a cell type-specific manner by breeding with mice that display cell type-restricted Cre recombinase expression. Due to the important role of DCs, mediating T cell tolerance under homeostatic conditions versus the induction of effector cells in response to infection, we evaluated this mouse model in a subline with CD11c-specific β2 integrin downregulation. We show here that splenic DCs derived from the studied mice showed enhanced cytokine production in response to stimulation with various toll-like receptor (TLR) ligands. In bone marrow-derived (BM) DCs used for subsequent analysis, stimulation resulted in the enhanced activation of signal transducers and activators of transcription (STAT) 1, 3 and 5, as well as the impaired expression of suppressor of cytokine signaling (SOCS) 2–6 proteins. However, β2-integrin-deficient BMDCs were also characterized by the attenuated expression of inflammation-associated gene sets. Correspondingly, in an experimental autoimmune encephalomyelitis (EAE) model, mice with a DC-specific β2 integrin knockdown were characterized by a delayed onset and attenuated course of disease. The spinal cords of these mice presented ex vivo with lower frequencies of T helper cell populations (Th)1/Th17. We conclude that our mouse model may prove to be a valuable tool to study the leukocyte-specific functions of β2 integrins in vivo under pathophysiological conditions.

## 2. Materials and Methods

### 2.1. Mice

Mice were bred and maintained in the Central Animal Facility of the Johannes Gutenberg-University Mainz under specific pathogen-free conditions on a standard diet according to the guidelines of the regional animal care committee. The “Guide for the Care and Use of Laboratory Animals, 8th edition” [37], as well as the 3R principles in laboratory animal experiments [38], were followed. Mice were sacrificed for organ retrieval according to § 4(3) TierSchG.

### 2.2. Generation of CD18^fl/fl^ Mice

A targeting vector based on plasmid BO44.2 was generated by PolyGene Transgenetics (Rümlang, Switzerland). The targeting construct comprises the sequence of exon 3 of the CD18 gene, contains a loxP–FRT–neo–FRT–loxP cassette, and is flanked by a 5′ short and a 3′ long arm of homology, the latter harboring the sequence of CD18 exon 4, 5 and 6. Vector and sequence information are given in Appendix A. Embryonic stem cells (clone JM8) derived from male agouti C57BL/6N blastocysts [39] were electroporated with double-digested (*NotI* + *SalI*) vector DNA and cultured in G418-containing selection media. The *ScaI*-digested genomic DNA (wild-type (WT) allele: 15 kb; targeted allele: 6.2 kb due to vector-mediated introduction of *ScaI*) of expanded clones was screened for the homologous integration of the targeting sequence by Southern blot using heat-denatured PCR products encompassing CD18 exon 1 (probe length: 569 bp; primers: sense, 5′-CAGTCCCCATCTCCACTCAG-3′ and anti-sense, 5′-GGCACTCTTTGAAGCACCAA-3′) and exon 7 (647 bp; 5′-ACACATGACAGCTGGGAAGA-3′ and 5′-GTCACCAACAGCGAACAGTT-3′) labeled via the incorporation of ^32^P-dCTP during PCR amplification (Ladderman Labelling Kit; TAKARA, Koto, Japan). With this approach, 2 out of 900 clones proved recombinant, and their karyotypes were evaluated. Afterwards, recombinant ES clones were injected into blastocysts that were subsequently transferred into the uterus of a recipient B6 Albino female mouse.

Chimeric offspring were initially selected according to the extent of agouti coat color. Strong male chimera were crossed back, over one generation, to the B6 Albino background. Agouti offspring were screened by PCR (542 bp; 5′-CAAGCTCTTCAGCAATATCACGGG-3′ and 5′-CCTGTCCGGTGCCCTGAATGAACT-3′) for the presence of the Neomycin (Neo) resistance gene. Neo^+^CD18^wt/fl^ chimera were crossed with Flp deleter mice (C57BL/6 background). Derived WT and Neo^−^CD18^wt/fl^ mice were differentiated by PCR (WT: 233 bp; Neo^−^CD18^wt/fl^: 487 bp; 5′-GTGACACTTTACTTGCGACCA-3′, 5′-TGCCAATAAAGAATTTCAGAGCC-3′). Heterozygous CD18^wt/fl^ mice were further bred to obtain homozygous CD18^fl/fl^ (B6.Cg-Itgb2^tm2.GrabS^) mice. The sequencing of a PCR-amplified (5′-GACCCCTAGATCTTCCCTGC-3′, 5′-ATAGAACCACCAACCTCGCA-3′) 1.37 kb fragment of genomic DNA encompassing the targeted region confirmed sequence integrity.

### 2.3. Generation of Mice with DC-Specific CD18 Deletion

CD18^fl/fl^ mice were bred with CD11c^Cre^ mice [40], and resulting CD18^wt/fl^CD11c^Cre+^ offspring were crossed back to the CD18^fl/fl^ background. Derived male CD18^fl/fl^CD11c^Cre+^ mice were paired with CD18^fl/fl^ females, yielding CD18^CD11c^ conditional knockout (cKO), termed hereafter as CD18^CD11c^ cKO and CD18^fl/fl^ CD11c^Cre-^ mice at same ratio. Since all mice were homozygous for the floxed allele, the offspring were screened by PCR for the presence of Cre recombinase only (Cre: 313 bp; 5′-ACT TGGCAGCTGTCTCCAAG-3′, 5′-GCGAACATCTTCAGGTTCTG-3′).

### 2.4. Immune Cell Isolation

Murine spleens were mashed with a sterile syringe plunger (Braun, Melsungen, Germany) on a 40 µm cell strainer (EASYstrainer™; Greiner Bio-One, Kremsmünster, Austria) pre-soaked with PBS. Mashed tissue was washed through the strainer using 10 mL of a buffer used for flow cytometric analysis (PBS 2% FCS, Pen/Strep, 2 mM EDTA) and spun down (1200 rpm, 10 min, 4 °C). The tissue of one lung was cut into small pieces and digested (37 °C, 45 min) using collagenase from *Clostridium histolyticum* type IA (Sigma-Aldrich, Deisenhofen, Germany) (0.5 mg/mL). Afterwards, tissue was further homogenized using a 10 mL syringe with a needle. The cell suspension was run through a 70 µm strainer and centrifuged (1400 rpm, 4 °C, 8 min). Erythrocytes were lysed using Gey’s lysis buffer. Ears were digested (400 U/mL collagenase IV, 1.25 mg/mL dispase, 100 U/mL hyaluronidase, 0.5 U/mL DNase, in RPMI) for 1 h at 37 °C in a thermo shaker. Subsequently, EDTA was added (10 mM) and incubated for an additional 5 min at 37 °C. Single cell suspensions derived from either tissue were passed through a 70 µm strainer, subsequently flashed with 10 mL of buffer (see above) to collect remaining cells, and pelleted (400 g, 7 min, 4 °C). For subsequent flow cytometric analysis, cells were resuspended in the buffer, counted, and incubated with antibodies (see Section 2.7). Splenic DCs were isolated with negative immune-magnetic selection using the Pan Dendritic Cell Isolation Kit as recommended by the manufacturer (Miltenyi Biotec, Bergisch Gladbach, Germany).

### 2.5. Bone Marrow-Derived Dendritic Cells (BMDC)

Bone marrow cells (2 × 10^5^/mL) were seeded either in 12-well cell cluster plates (1 mL) or in bacterial 10 cm dishes (10 mL) (both Greiner Bio-One) in an IMDM-based culture medium (5% FCS (PAN), 2 mM L-glutamine, 100 U/mL penicillin, 100 µg/mL streptomycin, 50 µM β-mercaptoethanol, and 10 ng/mL recombinant murine GM-CSF (Miltenyi Biotec). Culture media were replenished every 3 days of culture. BMDCs were subjected to experiments on days 10 or 11 of culture unless indicated otherwise.

### 2.6. Stimulation of DC

DC populations were stimulated overnight for the assessment of surface markers and cytokine production. For this, immunomagnetically negatively (Miltenyi Biotec) sorted splenic DCs (10^6^/mL) were resuspended in an RPMI medium supplemented with 5% FCS, 2 mM L-Glutamine, 0.1 mM nonessential amino acids, 50 μg/mL penicillin/streptomycin (all from Sigma-Aldrich, Deisenhofen, Germany), and 50 µM β-mercaptoethanol (Roth, Karlsruhe, Germany), and then they were seeded into 96-well microtiter plates (100 µL) (Greiner Bio-One, Frickenhausen, Germany).

BMDCs differentiated in 12-well cell cluster plates stimulated overnight were used for surface marker expression analysis (see Section 2.7). To assess cytokine concentrations (see Section 2.9), BMDCs differentiated in bacterial 10 cm dishes were stimulated in situ for immunofluorescence studies (see Section 2.10) or were harvested, resuspended in culture media (10^6^/mL), and seeded into 96-well microtiter plates (100 µL). All DC populations were stimulated with CpG ODN 1826 (Invivogen, San Diego, CA, USA), LPS (Sigma-Aldrich), and R848 (Invivogen) at the indicated concentrations. On the next day, cells were subjected to flow cytometric analysis and supernatants were retrieved for cytokine detection.

### 2.7. Flow Cytometry

Cells were washed in a staining buffer (PBS/2% FCS), and Fcγ receptors were blocked using a rat anti-mouse CD16/CD32 antibody (clone 2.4G2; 20 min). Then, samples were incubated (20 min, 4 °C) with fluorescence-labeled antibodies (Appendix A) as indicated. Antibodies were purchased from BD Biosciences (Franklin Lakes, NJ, USA), Thermo Fisher (Scientific, Waltham, MA, USA) or BioLegend (San Diego, CA, USA). After antibody incubation, samples were treated with fixable viability dye (FVD-eFl450 or -eFl780; Thermo Fisher) to delineate dead cells. The FoxP3-Staining Buffer Kit was used to detect intracellular transcription factors, as recommended by the manufacturer (Thermo Fisher). Cells were fixed and permeabilized for 45 min. Then, samples were incubated with antibodies for 30 min. Samples were analyzed with an Attune NxT flow cytometer equipped with 4 lasers (Thermo Fisher) and a FACSymphony with 5 lasers (BD Bioscience). Data were analyzed using FlowJo software (FLOWJO, Ashland, OR, USA).

### 2.8. Antigen Uptake and Processing

OVA-AF647 (antigen uptake) and OVA-DQ (antigen processing) (each 25 µg/mL) were applied to BMDCs differentiated in 12-well plates. Samples were incubated in parallel on ice and at 37 °C for 1 h. Samples were harvested and an MHCII-specific antibody was applied. The frequencies of MHCII-expressing cells that internalized and processed OVA were delineated by flow cytometric analysis.

### 2.9. Cytokine Detection

Cytokine contents were measured with flow cytometry (Cytometric Bead Array; BD Biosciences) and were analyzed using FCAP ArrayTM software v.1 (BD Biosciences) according to the instructions of the manufacturer.

### 2.10. Immunofluorescence

BMDCs (10^5^ in 100 µL PBS) were cytospun (4000 rpm, 5 min) onto microscope slides (Superfrost Plus, VWR; Darmstadt, Germany) using a Cytospin 3 (Thermo Fisher) and stored at −20 °C. The immunofluorescence detection of suppressor of cytokines (SOCS) proteins was carried out as described previously [41]. To this end, cytospins were first defrosted for 5 min and permeabilized by incubation with −20 °C methanol (Carl Roth, Karlsruhe, Germany) for 10 min. Subsequently, cytospins were washed twice with PBS/0.1% Tween (Carl Roth), followed by blocking unspecific binding sites with PBS/1% bovine serum albumin (BSA; Sigma-Aldrich, Deisenhofen, Germany)/10% mouse serum/10% goat serum (both from Dianova/Biozol, Hamburg, Germany) for 30 min at room temperature in a humified chamber. Subsequently, cytospins were incubated with rabbit anti-mouse polyclonal antibodies specific for SOCS2 (St. Johns Laboratory, London, UK), SOCS3 (Bioss, Woburn, MA, USA), SOCS4 (Sino Biologicals), SOCS5 (Bioss), and SOCS6 (Sino Biologicals) used at dilutions recommended by the respective manufacturer for 12 h at 4 °C in a humified chamber (50 µL PBS/1% BSA). In parallel settings, an isotype control was generated via incubation with a polyclonal rabbit IgG antibody (diluted 1:20; SouthernBiotech, Birmingham, AL, USA), whereas unstained controls were left untreated. Afterwards, samples were incubated with an Alexa Fluor 647-labeled goat-anti-rabbit monoclonal antibody (highly cross-adsorbed, diluted 1:500; Thermo Fisher) for 1 h in a humidified chamber. After an additional washing step, all specimens were incubated with Hoechst dye (100 µg/mL, 5 min) obtained from Thermo Fisher for nuclear staining. Then, samples were covered with a fluorescence mounting medium (DAKO; Agilent, Santa Clara, CA, USA). Immunofluorescence images of cytospins were taken using a TCS SP8 confocal laser scanning microscope (Leica, Mannheim, Germany). Images are depicted as the maximum projection of total z-stacks and were taken of the two representative regions of interest (ROI) of each cytospin slide, which included between 250 and 4300 cells/ROI. The brightness and contrast of images were adjusted in a homogenous manner.

Single-cell-based analyses were carried out using the Hoechst channel (blue) for the segmentation of cell nuclei in the open-source image analysis software QuPath (version 0.3.0) [42]), as described previously [43]. Segmentation was followed by the further classification of cells as positive or negative for a SOCS protein (SOCS-2, SOCS-3, SOCS-4, SOCS-5 and SOCS-6). The fluorescence intensity of SOCS protein expression by BMDCs was quantified in the Alexa 647-channel (*n* = 4–7 ROI per condition). The mean fluorescence intensities (MFI) of SOCS protein expression were calculated as the mean expression level in each ROI.

### 2.11. RNA Sequencing

Aliquots of BMDCs differentiated in 10 cm dishes were stimulated overnight with LPS (100 ng/mL). MHCII-positive cells were immunomagnetically isolated (Miltenyi Biotec) and lysed (RLT PlusLysis Buffer; Qiagen, Hilden, Germany). RNA was purified with the RNeasy Plus Micro Kit as recommended by the manufacturer (Qiagen) and quantified using a Qubit 2.0 fluorometer (Invitrogen). RNA quality was assessed with a Bioanalyzer 2100 using a RNA 6000 Pico chip (both from Agilent, Santa Clara, CA, USA). We used 10 ng of total RNA (integrity > 8) to generate barcoded an mRNA-seq cDNA library using the NEBNext^®^ Poly(A) mRNA Magnetic Isolation Module and NEBNext^®^ Ultra™ II RNA Library Prep Kit for Illumina^®^ (both from NEB, Ipswich, MA, USA), with a final amplification of 15 PCR cycles. RNA was quantified using the Qubit HS assay kit (Thermo Fisher), and the library size distribution was determined using the Bioanalyzer HS DNA assay (Agilent). Afterwards, barcoded RNA-Seq libraries were clustered using the HiSeq^®^ Rapid SR Cluster Kit v2 (Illumina, San Diego, CA, USA) using 8 pM, and 59 bps were sequenced on the Illumina HiSeq2500 using HiSeq^®^ Rapid SBS Kit v2 (59 cycles). Primary results were processed according to the Illumina standard protocol. Then, sequence reads were trimmed for adapter sequences and processed using the CLC Genomics Workbench software (v22.0, default settings; Qiagen). Reads were aligned to the GRCm39 genome. Total gene reads were exported to create a count table, and further data analysis was performed in RStudio (DESeq2 package). Differentially expressed genes were sorted based on the adjusted *p*-value, and a cut-off was set to 0.05. Results were illustrated using the R pheatmap package. GraphPad Prism 9 was used to create volcano plots of differentially expressed genes. The gene set enrichment analysis of normalized gene counts was performed using the GSEA 4.2.3 software (standard settings, gene set database: h.all.v7.5.1). A false discovery rate (FDR) *q*-value < 0.05 was considered statistically significant. Transcriptome data have been deposited in the GEO database, accession number GSE203448.

### 2.12. Real-Time PCR

Differentially pretreated splenic DCs were lysed in an RLT plus buffer (Qiagen), and total RNA was isolated using the RNeasy MiniPlus kit (Qiagen). cDNA was synthesized by applying the iScript kit (Bio-Rad, Munich, Germany). TNF-α (5′-CCACCACGCTCTTCTGTCTA-3′, 5′-AGGGTCTGGGCCATAGAACT-3′), IL-6 (5′-CCGGAGAGGAGACTTCACAG-3′, 5′-CAGAATTGCCATTGCACAAC-3′), SOCS2 (5′-AACCTGCGGATTGAGTACCA-3′, 5′-GGTACAGGTGAACAGTCCCA-3′) and SOCS4 (5′-TTCCCACCTCGCTCAGATTT-3′, 5′-GCTGGCCATTGGTATGTGAG-3′) were detected using the corresponding primer pairs. Ubiquitin C (5′-GTCTGCTGTGTGAGGACTGC-3′, 5′-CAGGGTGGACTCTTTCTGGA-3′) was used for normalization. All primers were obtained from Eurofins Scientific (Luxembourg City, Luxembourg). Reaction mixtures had a final volume of 25 µL and included 200 ng of cDNA, 70 nM of each primer, and 12.5 µL of 2× primaQUANT Master Mix high ROX (Steinbrenner Laborsysteme, Wiesenbach, Germany). Each sample was tested in duplicate. The thermal cycling conditions were 95 °C for 10 min, 40 cycles of 95 °C for 15 s, and 60 °C for 1 min, followed by a melting curve stage of 95 °C for 15 s and 60 °C for 1 min using an ABI 7300 real-time PCR cycler (Waltham, MA, USA).

### 2.13. Experimental Autoimmune Encephalomyelitis (EAE)

Mice were subcutaneously immunized at the tail base with 50 μg of MOG_35–55_ peptide (GenScript, Piscataway, NJ, USA) emulsified in complete Freund’s adjuvant (BD Biosciences). In addition, mice were intraperitoneally injected with 200 ng of pertussis toxin (List Biological Labs) on day 0 and day 2 post-immunization. Mice were observed daily to monitor body weight and EAE clinical symptoms on a scale from 0 to 5 as follows: 0—no disease; 1—flaccid tail; 2—impaired righting reflex and hind legs weakness; 3—complete hind legs paralysis; 4—complete paralysis of both hind legs with partial foreleg paralysis; and 5—moribund. To isolate cells from the spinal cord, the Multi Tissue Dissociation Kit 1 (Miltenyi Biotec) was used as recommended by the manufacturer. The cells were processed as described in the protocol and subsequently subjected to flow cytometric analysis.

### 2.14. Data Analysis

Data were analyzed using GraphPad PRISM v5.0 software (GraphPad Software Inc., San Diego, CA, USA). Statistical differences were assessed using a *t*-test in the case of comparisons of two groups or a one-way ANOVA and post-hoc Tukey’s test for post-hoc analyses in the case of multiple group comparisons, assuming significant differences at *p* < 0.05.

## 3. Results

### 3.1. Cre/Lox-Mediated Deletion of CD18 Exon 3 under Control of the CD11c Promoter Results in a Specific and Efficient Knockdown of β2 Integrins in DC

To enable the cell type-specific deletion of β2 integrins, pluripotent embryonic stem (ES) cells were transfected with a CD18 gene locus targeting vector, which was engineered to introduce loxP sites on either side of exon 3 and contained a neomycin (neo) resistance gene for the subsequent selection of transfectants (Figure 1A). Successfully transfected ES clones were injected into blastocytes that were implanted into foster mothers. Chimeric offspring were crossed with Flip recombinase (FLP) deleter mice to delete the neo cassette, thereby yielding CD18^fl/fl^ mice that were genotyped by PCR (Figure 1B). These mice were bred with CD11c^Cre^ mice that express Cre recombinase under the control of the CD11c gene promoter to enable DC-restricted Cre expression [40]. Through this process, CD18^CD11c^ cKO mice were obtained. The generation of CD18^fl/fl^ (Section 2.2) and CD18^CD11c^ cKO (Section 2.3) mice is described in more detail in the Materials And Methods section. The offspring were genotyped by PCR (Figure 1B).

Both in the blood (Figure 1C, left panel) and spleen (Figure 1C, right panel), the frequencies of most assessed leukocyte populations (B cells, T cells, polymorphonuclear granulocytes (PMNs), NK cells, monocytes, and macrophages (MACs)) remained unaltered in CD18^CD11c^ cKO mice and showed strong β2 integrin expression (Figure 1D). Similar observations were made in bone marrow (Appendix A). Only the frequency of cells expressing CD11c, the pan-DC marker in mice, was strongly downregulated in cells derived from the spleen (Figure 1C, right panel), as well as the ears and lungs of CD18^CD11c^ cKO mice (Figure 1E). These results indicate the successful knockdown of CD18 in DCs, as the surface expression of CD11c requires heterodimerization with CD18. However, all splenic DC populations (defined by the surface expression of CD317 for pDC, XCR-1 for cDC1, and CD172a for cDC2) remained unaltered (Figure 1F).

Altogether, these findings indicate that CD18^CD11c^ cKO mice are characterized by an efficient DC-specific knockdown of β2 integrins.

### 3.2. Downregulation of β2 Integrins in Splenic DCs Enhances Their Cytokine Production in Response to Stimulation

Splenic DC populations (cDC1, cDC2, and pDC) showed higher expressions of the costimulatory receptors CD80 and CD86 when spleen cells were stimulated with the TLR ligands LPS (TLR4) and R848 (TLR7), although below statistical significance in several cases (Figure 2A). Similar observations were also made for MHCII regulation (not shown). In contrast to the genotype-independent induction of marker expression, enriched splenic DCs derived from CD18^CD11c^ cKO mice showed the markedly stronger production of cytokines than DCs obtained from CD18^fl/fl^ mice, as exemplified for TNF-α, IL-6 and IL-10 (Figure 2B). Here, R848 showed a tendency to induce higher levels of cytokine expression than LPS and CpG used as an additional TLR ligand (TLR9). To elucidate the molecular base for the genotype-dependent differences in cytokine expression, we assessed the corresponding mRNA levels at short time points after the onset of stimulation with R848 (1 and 4 h) as the overall most potent cytokine-inducing agonist. In agreement with enhanced cytokine protein levels, the corresponding mRNA levels were somewhat higher in β2-integrin-deficient DCs, as shown for TNF-α (1 h) and IL-6 (4 h) after the onset of stimulation (Figure 2C), albeit below statistical significance. These observations suggest that increased cytokine levels in splenic DCs lacking β2 integrins may be due to elevated gene expression.

### 3.3. BMDCs Lacking β2 Integrins Present with a Somewhat Higher State of Activity, Associated with Elevated STAT Activity and Attenuated SOCS Expression

Due to the low frequency of primary DCs, we employed bone marrow-derived DCs (BMDCs) for subsequent analysis because, with regard to β2 integrins, express LFA-1, MAC-1, and CD11c/CD18, which is largely confined to DCs in mice.

BMDCs lacking β2 integrins showed the stronger induction of MHCII in response to stimulation with LPS, R848 and CpG than control BMDCs (Figure 3A, upper left panel). A similar tendency was observed for the costimulatory receptor CD40 in the case of LPS. A similar tendency was noted for R848, whereas CpG was not efficient (Figure 3A, upper right panel). The surface expression of CD80 and CD86 was enhanced to comparable extents in response to stimulation with either TLR ligand in a genotype-independent manner (Figure 3A, lower panel). Here, LPS was most potent, followed by CpG (not significant in the case of CD80 expression by CD18^fl/fl^ BMDC) and R848 (below significance in the case of CD80).

Furthermore, similar to primary splenic DCs, BMDCs devoid of β2 integrins showed a tendency to express some cytokines at higher levels (TNF-α: LPS; IL-1β: R848, CpG), albeit below statistical significance (Figure 3B). Since cytokine expression is largely regulated by STAT proteins, we assessed differences in genotype-dependent STAT activities. To this end, BMDCs were stimulated with LPS, which evoked the overall strongest stimulatory response. β2-integrin-deficient BMDCs were characterized by significantly higher levels of phosphorylated and thereby active STAT1, STAT3 and STAT5 upon stimulation with LPS (Figure 3C).

So far, our findings indicated that DCs with downregulated β2 integrin expression were characterized by exacerbated cytokine production and enhanced STAT activation. Cytokine production in leukocytes is known to be limited by SOCS proteins, which act on various levels, including the inhibition of STAT activation [46]. Therefore, we assessed genotype-dependent differences in SOCS expression. For this, BMDCs were stimulated overnight with LPS or R848, which yielded the strongest effects on cytokine induction in splenic DCs and BMDCs, respectively. SOCS protein levels were assessed by immunofluorescence using cytospins of unstimulated (Ctrl) and differentially stimulated BMDCs (Appendix A). Interestingly, the assessed SOCS proteins displayed significantly lower expression in unstimulated BMDCs (Ctrl condition), except for SOCS3 and SOCS4, which showed the same tendency though below significance (Figure 4A). Furthermore, whereas control BMDCs upregulated SOCS protein expression in response to stimulation by LPS and R848 in many cases to similar extents, BMDCs with attenuated β2 integrin expression were refractory in this regard. In response to stimulation with R848, which yielded similar SOCS protein induction to LPS, β2-integrin-deficient BMDCs showed somewhat lower induction of Socs mRNA upregulation than control BMDC, as shown for Socs2 and Socs4 (below statistical significance) (Figure 4B). However, under basal conditions, we observed no attenuated expression of either Socs mRNA in BMDCs lacking β2 integrins, in contrast to protein levels, which may indicate differences in mRNA translation efficacy or protein turnover.

Taken together, our results indicate that in DCs, β2 integrins are involved in the regulation of SOCS gene expression to limit stimulation-induced STAT activity and thereby cytokine production.

### 3.4. Knockdown of β2 Integrins in BMDCs Inhibits Induction of Genes Associated with Inflammatory Signaling

Next, we wanted to obtain a broader view on β2-integrin-dependent transcriptional changes in BMDCs. Under basal conditions, BMDCs with attenuated β2 integrin expression were characterized by marked differences in their gene expression profile compared to the control group, showing the differential regulation of a total of 573 genes (up: 517, Appendix A; down: 56, Appendix A), the top 50 of which are shown in Appendix A. Among the top 10 regulated genes, Wdfy1 (WD Repeat And FYVE Domain Containing 1)—which constitutes a positive regulator of TLR-induced signaling [47]—was strongly downregulated and Comm2 (COMM Domain Containing 2)—an inhibitor of NF-κB-dependent gene expression [48]—was highly upregulated (Appendix A). In broad accordance, gene enrichment analysis showed that the overall expression of genes involved in NF-κB-induced (TNF-α) signaling was lower in BMDCs lacking β2 integrins (Appendix A). On the contrary, genes encoding proteins required for gene expression and involved in metabolism were strongly upregulated (Appendix A). Along this line, gene enrichment analysis showed that genes that constitute targets of the transcription factors Myc and EF2, as well as genes involved in cell metabolism (mTORC1 signaling, fatty acid metabolism, and oxidative phosphorylation), were more expressed in BMDCs with impaired β2 integrin expression (Appendix A).

To assess the impact of β2 integrins on the gene expression of stimulated DCs, BMDCs were treated with LPS as the most potent TLR ligand. After overnight stimulation, BMDCs lacking β2 integrins expressed 185 genes to higher (Appendix A) and 15 genes to lower (Appendix A) extents than the corresponding control group (Figure 5A). Again, the positive regulator of TLR signaling Wdfy1 was expressed at lower level in β2-integrin-deficient BMDCs compared to the control group (Figure 5B). Additionally, thrombospondin 1 (Thbs1), which confers intercellular and cell–matrix interactions, was apparent at lower extent. On the contrary, genes associated with gene expression and metabolic processes were expressed at higher level in BMDCs lacking β2 integrins than in control BMDCs. Gene enrichment analysis revealed that stimulated BMDCs with attenuated β2 integrin expression were characterized by the impaired expression of genes that belong to several inflammatory pathways (Figure 5C). Similar to the unstimulated state, these BMDCs expressed EF2 and Myc target genes, as well as genes involved in oxidative signaling, at higher levels than control BMDCs.

### 3.5. DC-Specific Knockdown of β2 Integrin Inhibits Autoinflammatory Responses, Associated with Impaired Induction of T-Bet-Expressing T Effector Cells

So far, our results suggested that, on the one hand, increased cytokine production by DCs lacking β2 integrins due to impaired SOCS expression may exacerbate DC-mediated inflammation. On the other hand, transcriptome analysis indicated that these DCs may be less capable of exerting inflammatory processes. Therefore, we assessed the role of β2 integrins in DCs in vivo in the context of sustained inflammation. To this end, we employed EAE as a well-established autoimmune disease model of multiple sclerosis in which the immunization of mice with MOG peptide plus adjuvants results in the DC-mediated activation of autoreactive Th1/Th17 responses [49]. To rule out the idea that β2-integrin deficiency in DCs would interfere with the efficacy of immunization, we assessed antigen uptake and processing using BMDCs. As shown in Appendix A, BMDCs with the knockdown of β2 integrin actually displayed elevated levels of antigen uptake and processing.

We observed that CD18^CD11c^ cKO mice were characterized by a delayed and attenuated course of disease compared to CD18^fl/fl^ mice (Figure 6A). While the serum levels of proinflammatory TNF-α peaked 2 weeks after induction in a genotype-independent manner, IL-6 was only enhanced in immunized CD18^fl/fl^ mice (Figure 6B). At the end of the observation period, ex vivo analysis revealed that in inflamed spinal cords (SCs), the frequency of T-bet^+^ CD4^+^ T cells (Th1) was lower in CD18^CD11c^ cKO mice (Figure 6C). In the spleen (SP) and inguinal/paraaortic lymph nodes (LN), the frequencies of all Th populations (T-bet^+^ Th1, GATA3^+^ Th2, and RORgt^+^ Th17) and Foxp3^+^ regulatory T cells (Treg) were largely comparable. In the spinal cord, the frequencies of T-bet^+^RORγt^+^ and T-bet^+^FoxP3^+^ double-positive CD4^+^ T cell fractions were also attenuated (Figure 6D). Interestingly, however, the spinal cords of CD18^CD11c^ cKO mice contained more conventional cDC1 and cDC2, though the level of pDC remained unaltered (Figure 6E). It is noteworthy that in an ex vivo assay, less CD207^+^ DCs emigrated from FITC painted ears of CD18^CD11c^ cKO mice than that observed for the CD18^fl/fl^ control (Appendix A). Hematoxylin and eosin (H&E) stainings of spinal cords suggested somewhat lower leukocyte infiltration in the case of CD18^CD11c^ cKO mice (Appendix A).

Altogether, our results indicate that the β2 integrin knockdown in DCs resulted in stronger stimulation-induced activation, which was most pronounced on the cytokine level, due to attenuated SOCS expression. However, at the same time, these stimulated DCs were characterized by impaired T cell stimulatory activity, resulting in attenuated autoinflammatory reactions.

## 4. Discussion

The significance of β2 integrins, which are expressed by leukocytes in a cell-type specific manner [1], is highlighted by the phenotype of LAD1 patients. The molecular reasons for LAD1 are loss-of-function mutations of CD18, which results in the reduced expression of β2 integrins [26]. The extent of the impaired expression of functional of β2 integrins correlates with a predisposition towards recurrent, severe infections. The phenotype of LAD1 patients has been largely associated with the functional impairment of PMNs [27,34], which play a crucial role in innate defense towards pathogens [50]. So far, a number of mouse strains in which CD18 and thereby β2 integrins are expressed at very low levels (CD18^hypo^ [30]), are completely deficient (CD18^−/−^ [31]), or lack individual α subunits (CD11a–d [1]) have been established and were used by us and others for in vitro and in vivo studies. However, in either mouse model, all types of leukocytes are affected due to a constitutive knockout that hampers studies on the cell type-specific role of β2 integrins, especially in vivo. To overcome these limitations, we generated a conditional CD18 knockdown mouse model and reported on the outcome of a DC-specific β2 integrin knockdown.

Our in vitro results confirmed that β2 integrins may serve to limit DC activation, as reflected by enhanced cytokine production in the cKO cell line in response to stimulation. This finding is in broad accordance with a previous study by Yee and coworkers, who reported on elevated cytokine levels in CD18^−/−^ BMDC cultures in response to stimulation with various TLR ligands [51]. In the same study, similar results were obtained for bone marrow-derived macrophages. Here, CD18 deficiency was shown to result in a somewhat enhanced activity of NF-κB, which may result in the increased expression of proinflammatory mediators. Interestingly, CD11a- and CD11b-deficient BMDCs showed no hyperactivation. On the contrary, Ling et al. demonstrated that CD18^−/−^ BMDCs displayed significantly lower cytokine levels after stimulation compared to a WT control, which was associated with the lower phosphorylation of mitogen-activated protein kinases and IκBα in the former population [22]. Furthermore, whereas Yee and coworkers ruled out any contribution of CD11b to the hyperresponsiveness of BMDCs towards stimulation with different TLR ligands [51], Bai et al. demonstrated that the activation of CD11b-deficient BMDCs with TLR2 and TLR9, but not TLR4 ligands (here LPS), specifically promoted IL-12 release by downregulating miRNA 146a and thereby increasing the expression of its target Notch1 [52]. NOTCH1, in turn, acts as a stimulator of IL-12 expression [53].

The stimulation of DCs with danger signals such as TLR ligands univocally results in NF-κB activation, which is required to upregulate the expression of both surface markers and cytokines. Our observation of the somewhat elevated expression of activation markers, as well as increased cytokine production in β2-integrin-deficient BMDCs, may be explained by moderately enhanced NF-κB activity, as deduced from the results of Yee and coworkers [51]. The hyper-responsiveness of β2-integrin-deficient DCs was more pronounced on a cytokine level compared to surface marker expression, indicating a dysregulation of the STAT/SOCS signaling axis known to control primarily cytokine production [46,54]. The triggering of cytokine receptors, as well as TLR signaling [55], results in Janus kinase (JAK)-mediated phosphorylation and thereby the activation of STAT factors. STATs, in turn, enhance gene expression, especially of cytokines [56]. At the same time, STATs also upregulate the expression of SOCS proteins, which limit cytokine production by interfering with JAK-induced STAT activation [57]. Moreover, SOCS also attenuate TLR signaling [58] and NF-κB activity [59,60]. Therefore, our finding of elevated STAT activity and increased cytokine levels in β2-integrin-deficient DCs may be well-explained by attenuated SOCS protein levels. It is still unclear by which mechanism β2 integrins control SOCS expression. However, it has been demonstrated that upon binding to extracellular substrates, β2 integrins trigger the activation of focal adhesion kinase (FAK) [61], which was in turn demonstrated to promote, e.g., the expression of SOCS-3 [62]. It is conceivable that in the absence of β2 integrins, FAK activity might be strongly reduced and thereby negatively influence SOCS expression. Further studies are necessary to evaluate the potential role of this signaling cascade. Concerning the role of β2 integrins in STAT regulation, the treatment of NK cells with a MAC-1 agonist (leukadherin-1) was reported to impair STAT5 activity following cytokine stimulation. As a result, cytokine production by accordingly treated NK cells and monocytes was attenuated [63]. Furthermore, Yakubenko and coworkers demonstrated that the antibody-mediated activation of MAC-1 interfered with IL-13-mediated macrophage activation by blocking IL-4/IL-13 receptor signaling, as well as STAT-1, -3 and -6 phosphorylation [64]. The potential involvement of SOCS proteins in β2-integrin-regulated STAT activity has, however, not yet been studied.

As outlined above, β2 integrins may limit DC activation by promoting SOCS expression under basal conditions and early during stimulation. However, at the same time, β2 integrins may also be required for the expression of genes that confer a sustained activated state in DCs, as deduced from transcriptome analysis of CD18^CD11c^ cKO BMDCs. In this regard, we observed the strong downregulation of Wdfy1, which interacts with TLR3 and TLR4 on a protein level, and confers the activation of NF-κb and interferon regulatory factor (IRF) transcription factors as a signaling adaptor [47]. Moreover, β2-integrin-deficient BMDCs also overexpressed Commd2, which inhibits NF-κB activity by direct interaction [48]. Hence, it is conceivable that the dysregulated expression of Wydf1 and Commd2, which belonged to the group of most differentially expressed genes, may contribute to the overall impaired expression of inflammatory response gene sets in stimulated CD18^CD11c^ cKO BMDCs. So far, β2 integrins have not been associated with the expressional regulation of Wydf1 and Commd2. Further studies in our lab are intended to delineate which mechanisms β2 integrins in DCs control the expression of Wydf1 and Commd2, which role either factor plays in promoting the overall transcriptional alterations observed in β2-integrin-deficient DCs, and their role in shaping T cell responses.

In agreement with our observation of the impaired expression of inflammatory genes in β2-integrin-deficient BMDCs, we observed attenuated autoimmune reactions in EAE, characterized by the delayed induction of clinical symptoms and lower clinical scores. We showed that in the late phase of EAE (specifically in the spinal cord, as the main site of autoimmune inflammation), the frequencies of Th1, also comprising Th1/Th17 and Th1/Treg hybrid populations, were lowered in CD18^CD11c^ cKO mice. Accordingly, the spinal cords of CD18^CD11c^ cKO mice contained somewhat lower numbers of leukocytes, as deduced by H&E staining. EAE is known to mainly be driven by DC-induced autoantigen-specific Th1 and Th17 cell populations. [49]. Therefore, our findings suggest that in vivo, β2-integrin-deficient DCs have an impaired potential to activate (autoreactive) T cells. Considering the elevated uptake and processing of OVA antigens, as well as enhanced global cytokine production by stimulated β2-integrin-deficient DCs, it is less likely that the EAE-inducing immunization of mice with antigen peptides and adjuvants was less effective in CD18^CD11c^ cKO mice. A delayed onset and attenuated course of EAE has also been observed for CD11b^−/−^ mice [65], which suggests that CD11b-expressing cDC2 populations may play an important role in this regard. Furthermore, we observed a somewhat attenuated emigration of activated DCs from CD18^CD11c^ cKO skin explants. Similarly, murine DCs expressing mutated CD18, which was unable to interact with cytoskeletal Kindlin-3, were demonstrated to show an impaired migratory activity [66,67]. Our observation of enhanced spinal cord infiltration with cDC1 and cDC2 populations in the late phase of EAE in CD18^CD11c^ cKO mice could therefore be a consequence of the reduced migratory activity of β2-integrin-deficient DCs.

Moreover, it remains possible that the knockdown of β2 integrins as constituents of the immunological synapse may affect the interaction of DCs with T cells, resulting in attenuated T cell activation even in the case of the unaltered or elevated expression of MHCII and co-stimulators, as well as proinflammatory cytokines. We previously reported that LFA-1 [9] and MAC-1 [10] partly controlled the interaction of antigen-presenting DCs and syngenic antigen-specific CD4^+^ T cells. Of note, the inhibition of either β2 integrin on DCs resulted in elevated T cell proliferation, which suggested an overall inhibitory function of LFA-1 and MAC-1 on DCs. Interestingly, to the best of our knowledge, the role of CD11c/CD18 in this regard has not yet been studied. Anyway, it remains possible that the knockdown of all β2 integrins on DCs may yield an overall inhibitory effect due to overall dysregulated physical DC/T cell interactions. Furthermore, the dysregulated expression or activity of β2-integrin-regulated proteins (especially of the NF-κB signaling axis) may also play a role. Further studies are necessary to clarify these issues via the in depth analysis of DC/T cell interactions. In addition, ongoing studies are dedicated to elucidating which factors contribute to the attenuated course of EAE in CD18^CD11c^ cKO mice by assessing the extent of DC and T cell activation during the early priming phase after immunization and resulting immune responses prior to the onset and at the peak of clinical manifestations of the disease, including the assessment of T cell responses and the analysis of leukocyte infiltrations into spinal cord.

Ultimately, our mouse model allows for the efficient cell type-specific conditional knockdown of β2 integrins, which enables the study of the role of these multi-functional receptors in individual immune cell types under homeostatic and pathophysiological conditions in vivo.

## Figures and Tables

**Figure 1 cells-11-02188-f001:**
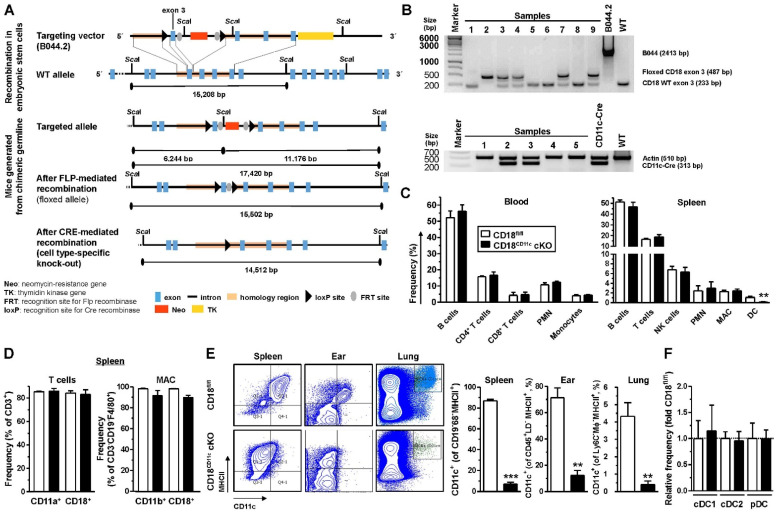
CD18^CD11c^ cKO mice display a DC-specific knockdown of β2 integrins. (**A**) Schematic overview of the generation of mice with a floxed CD18 exon 3 gene locus and the Cre recombinase-mediated knockout of exon 3. Sequence information of targeting vector B044.2 are given in Appendix A. (**B**) Examples of PCR-based genotyping of CD18^CD11c^ cKO mice assessing the presence of the floxed versus wild-type (WT) CD18 allele (upper panel) and Cre recombinase expression (lower panel). (**C**) Frequencies of leukocyte populations in blood (left panel; gating strategy shown in Appendix A) and spleen (right panel; gating strategy as described in [44]) were assessed by flow cytometry (mean ± SEM of 4–6 experiments). (**D**) Frequencies of β2 integrin α (CD11a and CD11b) and β (CD18) expressing splenic T cells (left panel) and macrophages (MACs) (mean ± SEM of 4 experiments). (**E**) Left panel: Dot plots show expression of MHCII and CD11c in cells derived from different organs and are representative of 3 experiments. Right panel: Quantification of CD11c-expressing cells in different organs (mean ± SEM of 3 experiments). (**F**) Relative frequency of splenic DC populations (gating strategy shown in Appendix A), normalized to the respective control (mean ± SEM of 4 experiments). (**C**,**E**) Statistical differences versus * CD18^fl/fl^ are indicated (unpaired *t*-test) (The asterisk denotes statistical differences between the “knockout” and the control group). ** *p* < 0.01, *** *p* < 0.001.

**Figure 2 cells-11-02188-f002:**
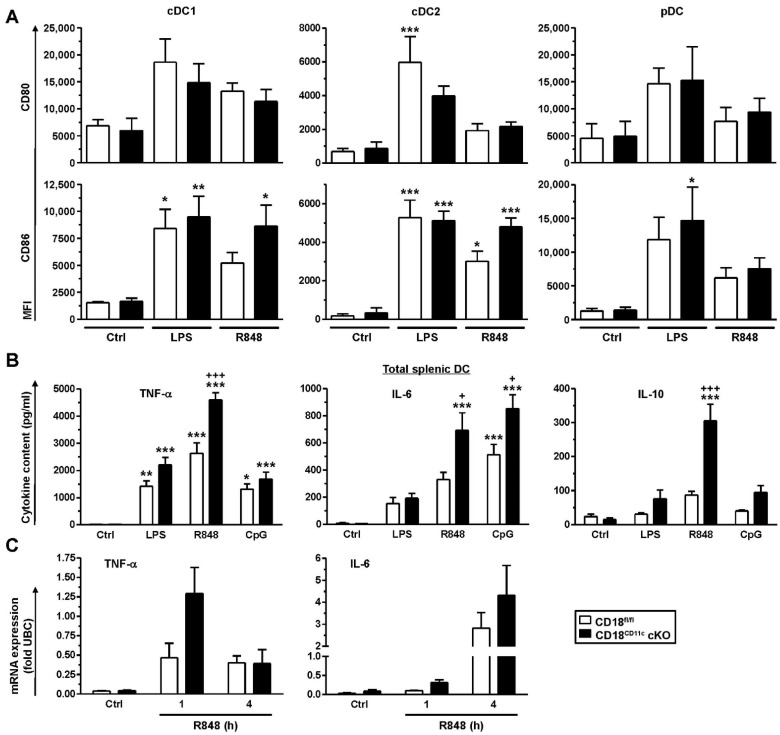
Downregulation of β2 integrins in splenic DCs results in overexpression of cytokines in response to stimulation. (**A**) Spleen cells were stimulated in parallel with LPS (100 ng/mL) and R848 (1 µg/mL) or left untreated (Ctrl). On the next day, expression of activation markers was assessed by flow cytometry (cDC1: XCR1^+^; cDC2: CD172a^+^; pDC: CD317^+^). Data denote the mean ± SEM of 4 experiments. (**B**) Immunomagnetically isolated splenic DCs were seeded into 96-well plates (10^5^/100 µL) and differentially stimulated (CpG: 250 ng/mL). On the following day, supernatants were retrieved for cytokine detection (mean ± SEM of 5 experiments). (**C**) Immunomagnetically sorted splenic DCs were stimulated with R848 for the indicated periods of time. mRNA levels were detected by real-time PCR and are given as fold of expression of the housekeeping gene ubiquitin C (UBC). Data denote the mean ± SEM of 3–4 experiments. (**A,B**) Statistical differences versus * CD18^fl/fl^ (Ctrl) and ^+^ CD18^fl/fl^ under same conditions are indicated (one-way ANOVA, Tukey test). *^,+^ *p* < 0.05, ** *p* < 0.01, ***^,+++^ *p* <0.001.

**Figure 3 cells-11-02188-f003:**
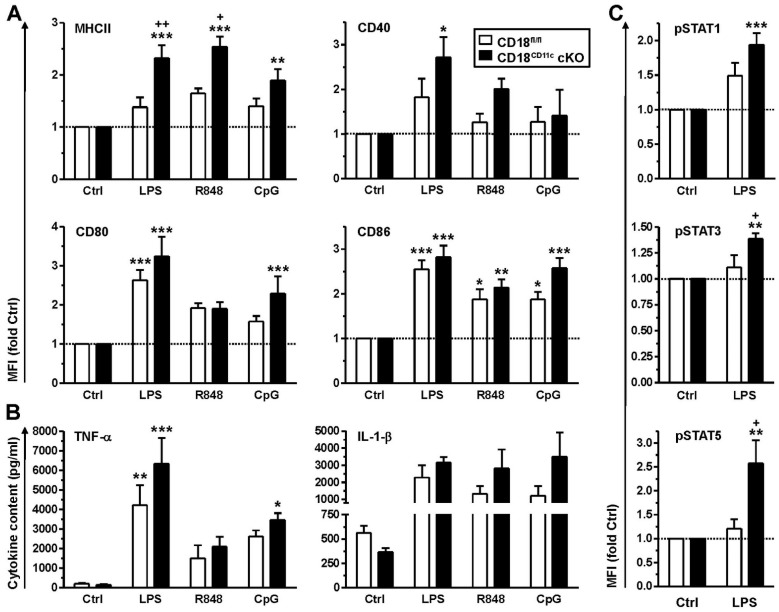
Bone marrow-derived DCs (BMDCs) with impaired β2 integrin expression show elevated expression of some activation markers and cytokines, accompanied by enhanced activity of STAT-1, -3 and -5. BMDCs were differentially stimulated overnight. (**A**) Expression of surface activation markers was assessed by flow cytometry (gating strategy as described in [45]). Data are given relative to the expression of unstimulated cells (Ctrl) (mean ± SEM of 9–12 experiments). (**B**) Cytokine levels were quantified by CBA (mean ± SEM of 3–8 experiments). (**C**) BMDCs were stimulated with LPS for 1 h. Levels of phosphorylated (p)STAT proteins were quantified by intracellular flow cytometry (mean MFI ± SEM of 5 experiments, given relative to Ctrl). (**A**–**C**) Statistical differences versus * CD18^fl/fl^ (Ctrl) and ^+^ CD18^fl/fl^ under same conditions are indicated (one-way ANOVA, Tukey test). *^,+^ *p* < 0.05, **^,++^ *p* < 0.01, *** *p* <0.001.

**Figure 4 cells-11-02188-f004:**
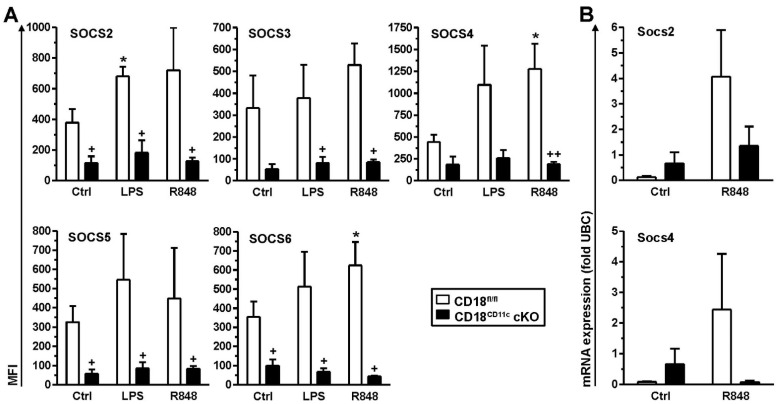
Impaired β2 integrin expression in BMDCs results in attenuated SOCS protein levels. BMDCs (CD18^fl/fl^ and CD18^CD11c^ cKO) were stimulated overnight in parallel with LPS and R848 or were left untreated (Ctrl). (**A**) On the next day, SOCS protein expression was assessed on cytospins by immunofluorescence. Representative graphs are shown in Appendix A. Data denote SOCS protein expression (mean MFI ± SEM of 3–4 experiments). (**B**) Expression of SOCS mRNA species was monitored 4 h after onset of stimulation by real-time PCR and is given in relation to the expression of the housekeeping gene UBC (mean ± SEM of 3–4 experiments). (**B**) Statistical differences versus * corresponding Ctrl and ^+^ CD18^fl/fl^ at corresponding conditions are indicated (one-way ANOVA, Tukey test). *^,+^ *p* < 0.05, ^++^ *p* < 0.01.

**Figure 5 cells-11-02188-f005:**
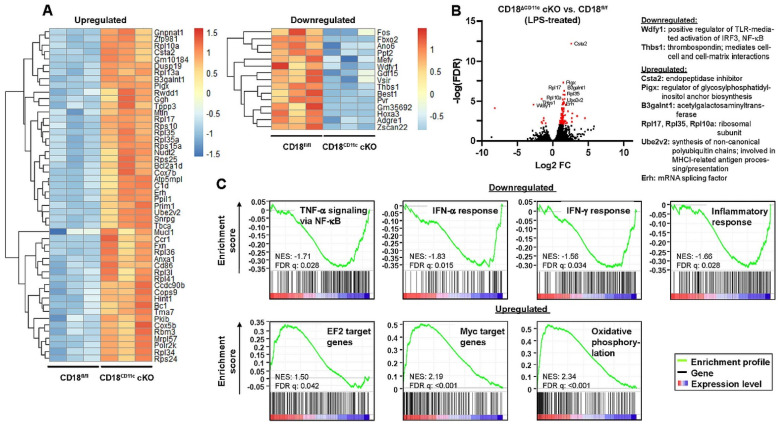
Downregulation of β2 integrins in BMDCs results in attenuated expression of genes associated with inflammatory signaling and upregulation of genes involved in metabolic pathways, as well as transcription factor Myc/EF2 targets. LPS-stimulated BMDCs (CD18^fl/fl^ and CD18^CD11c^ cKO, each *n* = 3) were subjected to RNA-seq analysis. (**A**) Heatmap representation of the top 50 significantly upregulated (left panel) and significantly downregulated (right panel) genes in CD18^CD11c^ versus CD18^fl/fl^ BMDCs (hierarchical clustering). The color legend denotes the level of gene expression (low: blue; high: red) and represents *z*-scores. (**B**) Volcano plot of all quantified mRNA species. Significantly regulated genes (*t*-test *q*-value < 0.05 and log2(fold-change) > 2) are given in red. The top 10 genes are named. (**C**) Gene set enrichment plots of significantly regulated pathways (FDR adjusted *q*-values < 0.05). The normalized enrichment score (NES) and FDR *q*-values are given.

**Figure 6 cells-11-02188-f006:**
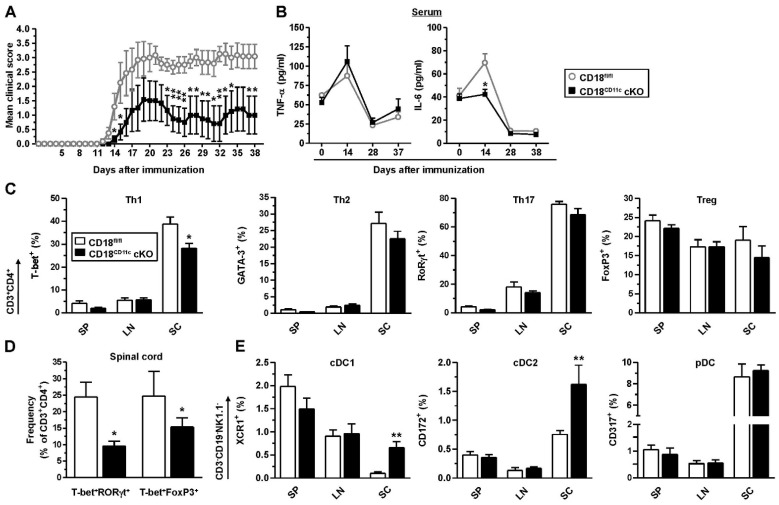
CD18^CD11c^ cKO mice display an attenuated course of EAE, associated with lowered frequencies of T-bet-expressing T helper cells in spinal cord. Mice were immunized on d 0 with MOG_35–55_ peptide in complete Freund’s adjuvant via the tail vein. Pertussis toxin was applied on d 0 and d 2. Mice were observed on a daily base for 38 d, followed by ex vivo analysis. (**A**) Clinical score of EAE symptoms (scoring details given in Section 2.13). Data denote the mean ± SEM of 7 mice per group, representative of 2 experiments. (**B**) Blood was retrieved throughout the observation period for cytokine detection (mean MFI ± SEM, *n* = 7 mice/group). (**C**) Frequencies of transcription factor-expressing CD3^+^CD4^+^ T cells in spleen (SP), inguinal/paraaortic lymph node (LN), and spinal cord (SC) (mean ± SEM, *n* = 3–7 mice/group) The gating strategy is depicted in Appendix A. (**D**) Frequencies of T-bet^+^RORγT^+^ and T-bet^+^Foxp3^+^ double-positive CD3^+^CD4^+^ T cells in spinal cord (mean MFI ± SEM, *n* = 3–4 mice/group). (**E**) Frequencies of DC populations in the various organs (mean ± SEM, *n* = 3–7 mice/group). The gating strategy is shown in Appendix A. (**A**–**D**) Statistical differences versus * CD18^fl/fl^ at corresponding conditions are indicated (one-way ANOVA, Tukey test). * *p* < 0.05, ** *p* < 0.01.

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
