# Peer review of "β2 Integrins on Dendritic Cells Modulate Cytokine Signaling and Inflammation-Associated Gene Expression, and Are Required for Induction of Autoimmune Encephalomyelitis"

_cells, 2022, doi:10.3390/cells11142188_

Round 1

Reviewer 1 Report

Heterodimeric β2 integrins are expressed by leukocytes which contributed to pathogen uptake, cell migration, immunological synapse formation and cell signaling. However, it is difficult to ascribe β2 integrins cell type-specific functions in vivo. In this manuscript, the authors generated a conditional CD18 knockdown mouse model and report on the outcome of a DC-specific integrin knockdown. Stimulated β2 integrin-deficient splenic DC showed enhanced cytokine production, but paradoxically, these BMDC showed attenuated expression of genes involved in inflammatory signaling, which showed a delayed onset and attenuated course of disease in EAE model. This model may be a valuable tool to study leukocyte-specific functions of β2 integrins in vivo. The idea is relative novel, however, the manuscript is desired to be revised due to some major and minor concerns as follows:

Major concerns:

The entire paragraph including lines 419-431 must be re-written to best elucidate the findings. 

As shown in Figure 4b, SOCS4 expression is also low in unstimulated BMDCs, which does not match the description of lines 424-425. And it seems contradictory about SOCS4 expression in Figure 4a and figure 4c. 

As showed in Figure 4c, there is no significant difference between group CD18fl/fl and CD18CD11c cKO in SOCS2 and SOCS4 mRNA expression, which is inconsistent with the attached text, similar questions also occurred in Figure 2c, please confirm.

4. The authors present incomplete sets of data in Figure 4a. Please show the representative immunofluorescence images of all groups in Figure 4a.

5. As shown in Figure S6, BMDC with a knockdown of β2 integrin displayed elevated levels of antigen uptake and processing, it is necessary to explore why the T cell stimulatory activity was impaired in β2 integrin-deficient DC, since the expression of MHC II has no difference in unstimulated BMDC (Figure 3a).

6. It is essential to show the histopathological features of MOG-induced inflammation in both groups (CD18fl/fl and CD18CD11c cKO) in section 3.5 to make these data visible and solid, including, but not limited to the histological staining of immune cells (such as CD4+T and macrophages) in EAE mice.

7. Figure 5 shows that the authors detected a large number of inflammation-related genes. Has the author picked out the more obvious ones for further research?

8. In line 399, CD40 is not mentioned above, please explain the reason for detecting CD40.

Minor concerns:

1. The sentence in lines 549-552 of the manuscript is easy to be confused. Please re-write these sentences for a better understanding.

2. Section “Materials and methods” was wrong numbered (2.6 occurred twice).

3. There are some typos in Lines 133-134, 5-GGCACTCTTTGAAGCACCAA3, and CAACAGCGAACAGTT 3 and Section “2.4. Immune cell isolation”.

Author Response

Major concerns:

The entire paragraph including lines 419-431 must be re-written to best elucidate the findings.

We have rewritten this paragraph to improve the description of the results.

As shown in Figure 4b, SOCS4 expression is also low in unstimulated BMDCs, which does not match the description of lines 424-425. And it seems contradictory about SOCS4 expression in Figure 4a and figure 4c.

We have corrected this mistake. Further, we mention and discuss the discrepancy between mRNA and protein expression levels („Interestingly, however, under basal conditions we observed no attenuated expression of either Socs mRNA in BMDC lacking b2 integrins, in contrast to protein levels, which may indicate differences in mRNA translation efficacy or protein turnover.“).

As showed in Figure 4c, there is no significant difference between group CD18fl/fl and CD18CD11c cKO in SOCS2 and SOCS4 mRNA expression, which is inconsistent with the attached text, similar questions also occurred in Figure 2c, please confirm.

We emphasize in the text that genotype-dependent differences in mRNA expression as shown in figures 2c and figures 4c show the same tendencies as observed on protein level, but were below statistical significance in eiter case.

  1. The authors present incomplete sets of data in Figure 4a. Please show the representative immunofluorescence images of all groups in Figure 4a.

We thank the reviewer for his hint. Due to the large number of graphs we show these in Figure S5.

  1. As shown in Figure S6, BMDC with a knockdown of β2 integrin displayed elevated levels of antigen uptake and processing, it is necessary to explore why the T cell stimulatory activity was impaired in β2 integrin-deficient DC, since the expression of MHC II has no difference in unstimulated BMDC (Figure 3a).

We thank the reviewer for this comment. We do not show in this manuscript that DC lacking ß2 integrins exert attenuated T cell stimulatory activity, but aim to clarify this issue in future experiments. We have included a paragraph in the discussion staiting that downregulation of ß2 integrins itself may have an impact on DC/T cell interaction, and thereby result in attenuated T cell activation despite enhanced expresison of MHCII, costimulatory receptors and proinflammatory cytokines. We state that further experiments are dedicated to elucidate this issue in detail.

  1. It is essential to show the histopathological features of MOG-induced inflammation in both groups (CD18fl/fl and CD18CD11c cKO) in section 3.5 to make these data visible and solid, including, but not limited to the histological staining of immune cells (such as CD4+T and macrophages) in EAE mice.

We also performed some histopathological analysis of spinal cord at the end of the experiment (day 38) and detected somewhat less leukocyte infiltration hematoxylin and eosin stained samples derived from ß2 integrin deficient mice, which is in accordance with the lower clinical score of these mice. We show these preliminary data in Figure S11. We will perform such analysis in new rounds of experiments aimed to decipher at earlier time points (pre-onset of clinical symptoms and at their peak) the functional defects of DC lacking ß2 integrins in terms of T cell activation (extent, polarization) and the associated infiltration of leukocyte populations. However, we are still waiting for the permission to perform these additional in vivo experiments, and therefore can not provide according results within a short period of time. We have included an according statement in the discussion section.

  1. Figure 5 shows that the authors detected a large number of inflammation-related genes. Has the author picked out the more obvious ones for further research?

We clarify in the discussion section that ongoing studies in our lab are dedicated to elucidate the relation between ß2 integrins and the genes Wydf1 and Commd2. Both genes regulate NF-kB signaling and are differentially regulated in BMDC lacking b2 integrins („Further studies in our lab are intended to delineate by which mechanisms ß2 integrins in DC control the expression of Wydf1 and Commd2, and which role either factor plays in promoting the overall transcriptional alterations observed in ß2 integrin-deficient DC, and their role in shaping T cell responses.“).

  1. In line 399, CD40 is not mentioned above, please explain the reason for detecting CD40.

We extended the according sentence clariying that CD40 constitutes a costimulatory receptor („A similar tendency was observed for the costimulatory receptor CD40“).

Minor concerns:

  1. The sentence in lines 549-552 of the manuscript is easy to be confused. Please re-write these sentences for a better understanding.

We have rewritten this sentence.

  1. Section “Materials and methods” was wrong numbered (2.6 occurred twice).

We have corrected this mistake.

  1. There are some typos in “Lines 133-134, 5’-GGCACTCTTTGAAGCACCAA’3, and CAACAGCGAACAGTT ‘3” and Section “2.4. Immune cell isolation”.

We have corrected these mistakes.

Reviewer 2 Report

Reviewer Comments to Author

The paper by Bednarczyk et al. reports the generation and characterization of a conditional CD18 knockdown mouse model and describes the outcome of a DC-specific beta2 integrin knockdown. The mouse model established could be useful to study beta2 integrin leukocyte-specific functions in vivo. The topic is very interesting and the results presented are valuable. In my opinion this paper is scientifically accurate and well written. The quality of English is good.

Nevertheless, some points of criticism emerge, and major revision are needed before the manuscript can be accepted for publication.

Major points:

The results described are quite complex and often described too concisely; the results should be better detailed and clarified.

Due to the lacking of statistical significance, some results should be described taking that into consideration, as if there is no statistical difference, the authors could not state there is or isn’t an increase/decrease in the measurement/effect considered. Please, rephrase.

The authors should better clarify why for some experiments they evaluated the effect of both R848 and LPS, while in others they showed the results deriving from the treatments with only one of the two TLR activators.

Minor points:

-page 9, lines 374-375: is there any statistical difference among groups in figure 2C? Otherwise, the sentence should be modified. And why did the authors test only the effect of R848? What about LPS effects?

-lines 398-400: the description of the results should be better detailed, as for example R848 and LPS showed different effects on BMDC (Figure 3a).

-In figure 3 some graphs lack label of y axis.

-Figure 3c: is there a statistical difference between the levels of pSTAT1 in BMDC stimulated with LPS, deriving from CD18fl/fl and CD18CD11c cKO?

-line 424: “with the exception of SOCS4” refers only to LPS-treated BMDC. Please, rephrase.

-lines 428-430 and figure 4c: if there is no statistical difference, then it wouldn’t be possible to state that there is a reduction. Please, rephrase.

-Figure 4: The types of data shown in figure 4 are confusing: why did the authors showed only the results deriving from different techniques only for one/two SOCS proteins? It would better to have the complete panel of experimental data for all the SOCS proteins.

-line 446: rephrase this sentence as on the basis of the results shown until here, the authors cannot state that beta2 integrin promotes SOCSs expression.

-line 470: I would suggest to rephrase as: “After LPS-overnight stimulation…”

-line 505: delete the word “according”

-Typos in caption of figure S1:

-line 318: structure;

-line 322: cloned;

-line 324: upstream

Author Response

Major points:

The results described are quite complex and often described too concisely; the results should be better detailed and clarified.

We have thoroughly revised the results part to describe results in more detail and to enhance clarity of presentation.

Due to the lacking of statistical significance, some results should be described taking that into consideration, as if there is no statistical difference, the authors could not state there is or isn’t an increase/decrease in the measurement/effect considered. Please, rephrase.

We have rewritten these passages.

The authors should better clarify why for some experiments they evaluated the effect of both R848 and LPS, while in others they showed the results deriving from the treatments with only one of the two TLR activators.

We have included statements explaining in each case the reasons for chosing the according TLR ligand(s), which most often depended on the outcome of the previous experiment.

Minor points:

-page 9, lines 374-375: is there any statistical difference among groups in figure 2C? Otherwise, the sentence should be modified. And why did the authors test only the effect of R848? What about LPS effects?

We thank the reviewer for this comment. There are no significant difference on mRNA level (figure 2c). We have altered the description of the result to indicate so more clearly („…, albeit below statistical significance.“). We also added a statement explaining that we used only R848 as a stimulus since this agent induced stronger cytokine responses than LPS (and CpG) („…with R848 (1h, 4h) as the overall most potent cytokine inducing agent“). Therefore, LPS effects were not tested.

-lines 398-400: the description of the results should be better detailed, as for example R848 and LPS showed different effects on BMDC (Figure 3a).

We have extended the description of the results, focusing on agonist-dependent differences in the simulatory response.

-In figure 3 some graphs lack label of y axis.

We have corrected these mistakes in figure 3 and also in figures 3 and 4.

-Figure 3c: is there a statistical difference between the levels of pSTAT1 in BMDC stimulated with LPS, deriving from CD18fl/fl and CD18CD11c cKO?

We observed no statistical significant difference between these groups (p=0.1137).

-line 424: “with the exception of SOCS4” refers only to LPS-treated BMDC. Please, rephrase.

We rephrased this sentence („Of note, with the exception of SOCS4 all other assessed SOCS proteins displayed significantly lower expression in unstimulated BMDC (Ctrl condition)“).

-lines 428-430 and figure 4c: if there is no statistical difference, then it wouldn’t be possible to state that there is a reduction. Please, rephrase.

We thank the reviewer for this comment. We have altered this statement accordingly („In agreement with attenuated induction of SOCS protein expression in response to stimulation, upregulation of SOCS mRNA levels in ß2 integrin-deficient BMDC upon stimulation was somewhat lower, as assessed for Socs2 and Socs4“).

-Figure 4: The types of data shown in figure 4 are confusing: why did the authors showed only the results deriving from different techniques only for one/two SOCS proteins? It would better to have the complete panel of experimental data for all the SOCS proteins.

We thank the reviewer for this hint. We show representative immunofluorescence pictures in figure S5.

-line 446: rephrase this sentence as on the basis of the results shown until here, the authors cannot state that beta2 integrin promotes SOCSs expression.

We agree with the reviewer. We have rephrased this sentence accordingly (b2 integrins are involved in the regulation of SOCS expression).

-line 470: I would suggest to rephrase as: “After LPS-overnight stimulation…”

We have rephrased this sentence.

-line 505: delete the word “according”

We have deleted this term.

-Typos in caption of figure S1:

-line 318: structure;

We have corrected this mistake.

-line 322: cloned;

We have corrected this mistake.

-line 324: upstream

We have corrected this mistake.

Round 2

Reviewer 1 Report

The authors carried out the reviewer's comments. However, few concerns about the manuscript need to be rephrased:

1. Section “Materials and methods” was wrong numbered (2.7 occurred twice),Please reorder all numbers in this section.

2. -line 450: rephrase as: “with LPS or R848.

3. -line 464: delete the word “for”.

4. -line 500: rephrase as: “(Figure S6b).

Author Response

  1. Section “Materials and methods” was wrong numbered (2.7 occurred twice),Please reorder all numbers in this section.

We renumbered all sections in the materials and methods section and  corrected the numbering o according references throughout the manuscript and in the supplementals.

  1. -line 450: rephrase as: “with LPS or R848”.

We rephased the sentence as suggested.

  1. -line 464: delete the word “for”.

We deleted that term as suggested.

  1. -line 500: rephrase as: “(Figure S6b)”

We corrected this mistake.

All alterations are highlighted in yellow.

We thank the reviewer for his effort.

Reviewer 2 Report

In my opinion, authors completely fulfilled major and minor revisions requested.

Author Response

We thank the reviewer for his comments and suggestions to improve the manuscript.